# When should lockdown be implemented? Devising cost-effective strategies for managing epidemics amid vaccine uncertainty

**Nathan J. Doyle**[1,2]*, **Fergus Cumming**[3], **Robin N. Thompson**[4], **Michael J. Tildesley**[2]

**1** EPSRC Centre for Doctoral Training in Mathematics for Real-World Systems, Mathematics Institute, University of Warwick, Coventry, United Kingdom, **2** The Zeeman Institute for Systems Biology & Infectious Disease Epidemiology Research, Mathematics Institute and School of Life Sciences, University of Warwick, Coventry, United Kingdom, **3** Foreign, Commonwealth and Development Office, London, United Kingdom, **4** Mathematical Institute, University of Oxford, Oxford, United Kingdom

\* nathan.doyle@warwick.ac.uk

**Data Availability Statement:** All data utilised in this study are publicly available, with relevant references and data repositories provided in the following github repository: https://github.com/

## Abstract

During an infectious disease outbreak, public health policy makers are tasked with strategically implementing interventions whilst balancing competing objectives. To provide a quantitative framework that can be used to guide these decisions, it is helpful to devise a clear and specific objective function that can be evaluated to determine the optimal outbreak response. In this study, we have developed a mathematical modelling framework representing outbreaks of a novel emerging pathogen for which non-pharmaceutical interventions (NPIs) are imposed or removed based on thresholds for hospital occupancy. These thresholds are set at different levels to define four unique strategies for disease control. We illustrate that the optimal intervention strategy is contingent on the choice of objective function. Specifically, the optimal strategy depends on the extent to which policy makers prioritise reducing health costs due to infection over the costs associated with maintaining interventions. Motivated by the scenario early in the COVID-19 pandemic, we incorporate the development of a vaccine into our modelling framework and demonstrate that a policy maker's belief about when a vaccine will become available in future, and its eventual coverage (and/or effectiveness), affects the optimal strategy to adopt early in the outbreak. Furthermore, we show how uncertainty in these quantities can be accounted for when deciding which interventions to introduce. This research highlights the benefits of policy makers being explicit about the precise objectives of introducing interventions.

## Author summary

While often necessary to contain an infectious disease outbreak, extensive interventions result in education, economic and societal harms. For the optimal response to be determined, the trade-off between the costs and benefits of disease control requires policy makers to define and communicate their objectives for managing the outbreak. We use a mathematical model to simulate outbreaks where social distancing is implemented when

ndoyle815/Epidemic-strategies-amid-vaccine-uncertainty.

**Funding:** N.J.D. and M.J.T. were supported by the Engineering and Physical Sciences Research Council through the Mathematics of Systems II Centre for Doctoral Training at the University of Warwick (grant number EP/S022244/1). The funders had no role in study design, data collection and analysis, decision to publish, or preparation of the manuscript.

**Competing interests:** The authors have declared that no competing interests exist.

disease prevalence within hospital settings is high. We consider four distinct strategies for managing the outbreak based on different hospital prevalence thresholds for switching between levels of social distancing. The optimal strategy to implement depends on how a policy maker balances the importance of reducing the number of hospitalisations and the costs of maintaining interventions. The optimal strategy to implement at the beginning of the outbreak is further impacted by beliefs regarding the future availability of a vaccine. We develop a quantitative decision making framework which explicitly considers the objectives of policy makers and allows robust strategies for future disease outbreaks to be designed.

## Introduction

The COVID-19 pandemic has highlighted the importance of mathematical and statistical models for informing and guiding public health policy decisions [1–3]. For example, infectious disease models can be used to assess the effects of non-pharmaceutical interventions (NPIs) [4] or to test different vaccination strategies [5] in real-time during an epidemic, informing policy makers about effective interventions to implement moving forwards. Policy makers are required to consider a range of costs in order to determine the optimal policy for outbreak management. The overall cost of an epidemic includes both the direct effects of disease (as quantified by public health metrics such as the total number of infections, hospitalisations and deaths) and the economic costs associated with the maintenance of interventions designed to suppress the epidemic. It is usually desirable to take both measures into account when deciding upon the optimal strategy to implement, but the degree to which multiple objectives relating to disease burden and economic factors are weighed in the final decision is often unclear. In the early stages of the COVID-19 pandemic, for example, there was extensive discussion regarding the contrast in the timing and principles of intervention policies adopted in different countries [6]. Certain governments such as China and Australia favoured a "zero-COVID" suppression strategy as opposed to less stringent policies adopted by countries such as Sweden. In the absence of a vaccine, countries only had NPIs at their disposal to mitigate disease spread. These intensive interventions were effective at controlling the epidemic [7], but resulted in negative economic and social consequences [8, 9]. On the other hand, less stringent approaches reduce the non-disease harms, but result in more individuals being exposed to infection and hence more severe disease outcomes in the short term, as observed in Sweden and the United Kingdom where less intense restrictions were adopted to curb COVID-19 in summer 2020 [10]. For models to be used to identify the optimal policy, it is necessary to define the objectives of disease control explicitly [11]. If policy objectives are clearly stated, epidemiological modellers can construct an objective function that can be quantified and used to guide decision making.

Objective functions have been constructed previously in epidemiological studies in which optimal control theory is applied to identify optimal disease management strategies [12–17]. Factors typically considered in the cost function include a measure of disease burden and time-dependent control variables, balanced by weighting coefficients. The disease burden contribution can be based on quantities such as the number of infected individuals [14] or can instead be based on the impact on healthcare resources (through e.g., the number of hospitalised individuals). The cost of control can include treatment and vaccination effort [16], which is often included in a nonlinear fashion to reflect, for example, the disproportionate costs at heightened control efforts [17]. Once an objective function has been formulated, different

control interventions can be tested. In some studies, a limited range of different interventions are tested, and in others it is possible to identify the optimal strategy from a substantial range of possibilities.

In a previous study, Probert *et al.* [11] discussed the need for an objective function to be constructed for evaluating intervention strategies in the context of foot-and-mouth disease (FMD). Strategies for FMD control can incorporate the targeted culling of livestock, including those that are not yet infected, which leads to both the death of animal hosts and substantial direct economic costs to farmers. Those authors demonstrated that the optimal action is highly dependent upon both the objectives of disease management and the epidemiological model used to guide decision making. In that study, different strategies are ranked comparatively rather than highlighting the best performing strategy alone. This is important as it can inform policy makers about robust strategies for disease control that perform relatively well across a range of modelling assumptions and objective functions. If the cost function is well-defined, policy makers may consider the risk of a catastrophic cost occurring (i.e., a reasonable worst case scenario), which can be explored if the underlying simulation model incorporates uncertainty. We build on this previous work by considering how different objective functions lead to different optimal interventions in the context of a novel disease of humans, accounting for features of outbreaks that are particularly pertinent to outbreaks in human populations (e.g. the rapid development of a vaccine, as was undertaken during the COVID-19 pandemic).

A key challenge for epidemiological modellers when designing models to guide interventions is to account for uncertainties about outbreak dynamics and the effects of those interventions, particularly in the early stages of the outbreak. Such uncertainties should be reflected when evaluating interventions using any objective function. Examples of sources of uncertainty which may affect the projected impacts of the outbreak include uncertainty in the parameters underlying pathogen transmission [18], the effectiveness of a potential vaccine and/or other interventions [19] and the economic impacts of interventions [20]. A particular source of uncertainty which likely affected decision making at the beginning of the COVID-19 pandemic was related to future vaccination prospects. COVID-19 vaccines were developed rapidly as vaccination became a global research priority early in the pandemic [21]. However, the time that it would take for effective vaccines to be developed and approved, as well as their eventual uptake, were unknown in the early months of the pandemic [22]. This likely affected early decision making regarding the implementation of NPIs, as the swift arrival of a highly effective vaccine would allow policy makers to maintain interventions with assurance of an imminent path to higher levels of (vaccine-induced) immunity in the population. Thus, a key feature of our current study is an investigation into how the optimal policy to adopt early in an outbreak changes based on uncertainty about whether or when a vaccine will be developed, as well as its eventual effectiveness and coverage.

In this paper, we present an age-structured mathematical model for the spread of a novel emerging pathogen. The model explicitly tracks the number of individuals in hospital as a result of severe infection during the epidemic, which is used as a trigger to switch between three levels of population-wide infection control, which we refer to hereafter by "control states". We consider four different possible strategies for disease control, each of which is defined by the precise trigger values at which control states are changed. The performance of each strategy is evaluated using model projections and an objective function that takes into account disease burden, the cost of maintaining interventions and the risk of hospital capacity being exceeded at any stage during the epidemic. Crucially, we demonstrate how the optimal strategy changes according to the relative weighting of disease burden and intervention stringency contributions in the objective function, as well as the time horizon that is considered.

Our goal is to develop a general epidemiological modelling framework that can be extended and used to guide interventions during future outbreaks of a range of pathogens, rather than to provide a detailed model that represents specific features of the COVID-19 pandemic. However, to demonstrate how the model can be used in specific scenarios, we consider a situation in which NPIs are being implemented but a vaccine may become available in future (as was the case early in the COVID-19 pandemic). We show how uncertainty in the time at which the vaccine will become available can be accounted for when determining the current optimal strategy. We show that the optimal strategy can be different when accounting for the possibility of vaccine development compared to the vaccine-absent scenario. Finally, we consider different ways in which objective functions can be used to identify optimal interventions—including minimising the expected value of the objective function or minimising different quantiles of the objective function. The latter approach allows policy makers to choose, for example, to introduce interventions to limit impacts of the "worst case" scenario. Taken together, this work provides a framework that can be used to test different interventions during future outbreaks of a range of pathogens.

## Results

### Model simulations

First, we explored the effects of the four strategies considered on outbreak dynamics. Simulations were performed by numerically solving the differential equation system (see supplementary material S1 Text), checking each day whether a switching threshold $T_{ij}$ is triggered and implementing policy decisions accordingly. Each simulation was preceded by a 30-day preliminary simulation without any intervention, since in reality interventions are not implemented instantaneously when a pathogen first arrives in the population.

Without vaccination, each strategy considered led to different outbreak dynamics (Fig 1). Hospital levels in the Cautious easing strategy rose to a high first peak as a consequence of the high $T_{12}$ threshold. This strategy had the fewest waves of infection across the time frame used and was also the only strategy in which Lockdown was only implemented once. In contrast,

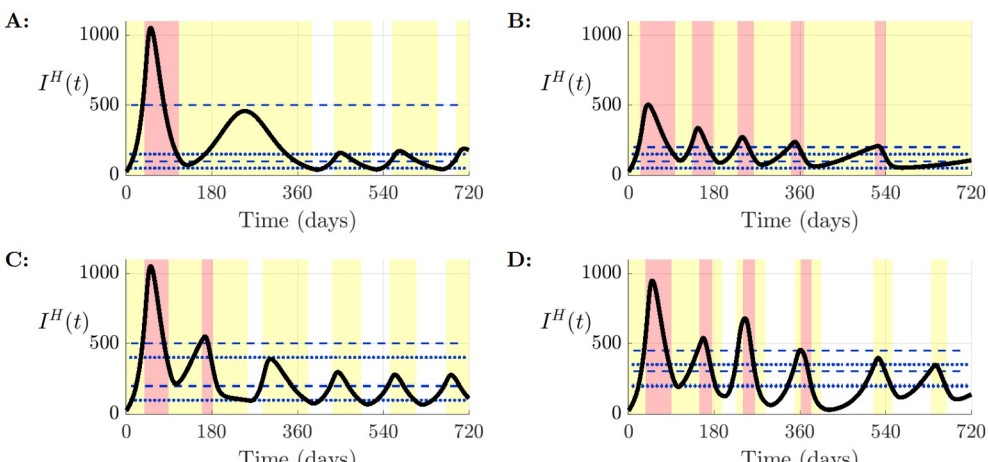

**Fig 1. Hospitalisation dynamics in the absence of vaccination.** Number of active hospitalised individuals across all age cohorts $I^H(t)$ for each strategy, in the scenario with no vaccination. Shaded are the periods spent in (red) Lockdown, (yellow) Partial Lockdown and (white) Inaction. Blue dashed lines represent the reintroduction thresholds (where $T_{12} > T_{01}$), and blue dotted lines represent the relaxation thresholds (where $T_{21} > T_{10}$). Strategies are labelled as A: S1 (Cautious easing), B: S2 (Suppression), C: S3 (Slow control) and D: S4 (Rapid control).

under the Suppression strategy, the narrow gap between thresholds led to frequent switching between Partial Lockdown and Lockdown without a break in measures to return to Inaction. However, high stringency was rewarded by generally low prevalence and overall disease burden was lowest when measured throughout the epidemic. For the Slow control strategy, there were two short successive periods of Lockdown in response to high levels of infection. This resulted in the effective reproduction number falling below one after the second lockdown and low hospital occupancy was maintained before a brief return to Inaction and smaller waves of infection. Finally, the narrow gaps between switching thresholds for the Rapid control strategy allowed the epidemic to be managed at a moderate but sufficiently low level of infection such that hospital capacity was never overwhelmed. This required regular switching between control states—this simulation moved through the largest number of phases of all strategies with four phases of Lockdown in total. Additionally, the high $T_{10}$ threshold meant that this strategy allowed the longest and most frequent periods of Inaction.

Outbreak dynamics with vaccination are shown in S5 Fig for initial vaccine deployment time $T = 360$ and eventual coverage (or effectiveness) $\eta = 0.9$. Since the vaccination coverage was sufficiently above the critical level $1 - 1/R_0$ required to eradicate infection [23], and waning immunity was assumed to occur slowly, the vaccine presented itself as a rapid pathway out of the epidemic for this scenario, with the pathogen approaching elimination after the onset of vaccination. However, there remained the possibility for smaller waves of infection, and corresponding reintroduction of NPIs, while vaccination was taking place.

The ranking of strategies using the objective function (1) depends upon the policy maker's choice about the relative importance of reducing the costs due to disease or the costs of stringent public health measures. In our model, this choice determines the value of the parameter $w$. For high values of $w$, the policy maker favours the reduction of cumulative hospitalisations in the objective function, whereas for low values of $w$ their priority switches to reducing the cost of maintaining interventions (see objective function (1)). The ranking of the four strategies is shown for a range of values of $w$, both in the absence of and with vaccination, in Fig 2. When a short time horizon $t_f$ is considered, the optimal strategy was either Slow Control or Suppression, depending on the value of $w$. Over longer time horizons, each of the four strategies were optimal in different scenarios, again depending on the precise value of $w$ but also the level of hospital capacity. When hospital capacity $H_c$ is lowered from 1,250 to 1,000 individuals, the Cautious Easing and Slow Control strategies are suboptimal as their first wave peaks (1,052 individuals) would overwhelm capacity and are thus infeasible strategies (Fig 2D). To allow for the consideration of all four strategies henceforth, $H_c$ will remain fixed at 1,250 individuals (1.25 beds per 1,000 population). The variance in strategy rankings across Fig 2 demonstrates that the objective of the policy maker is crucial in determining the optimal strategy.

## Incorporating vaccination uncertainty

We then simulated the model while varying the vaccine arrival time and coverage parameters $(T, \eta) \in \Psi$ to represent plausible vaccination scenarios. The time horizon used in each model simulation was 1,080 days (approximately 3 years). Surface plots depicting the overall cost of the outbreak under each strategy as a function of $w$ and vaccine arrival time $T$ are shown in Fig 3. Different rows in Fig 3 reflect different levels of eventual vaccine coverage, $\eta$. Analogous surface plots of cost by strategy are also provided in which the vaccine coverage parameter $\eta$ is varied continuously for specific vaccine arrival times $T = 360$, $T = 630$ and $T = 900$ (S6 Fig). Notably, the most stringent strategy (Suppression) is most likely to be optimal when the policy maker prioritises lowering disease costs (i.e. a high value of $w$) and when the vaccine is developed quickly and high vaccination coverage is achieved (see, for example, low values of $T$ in

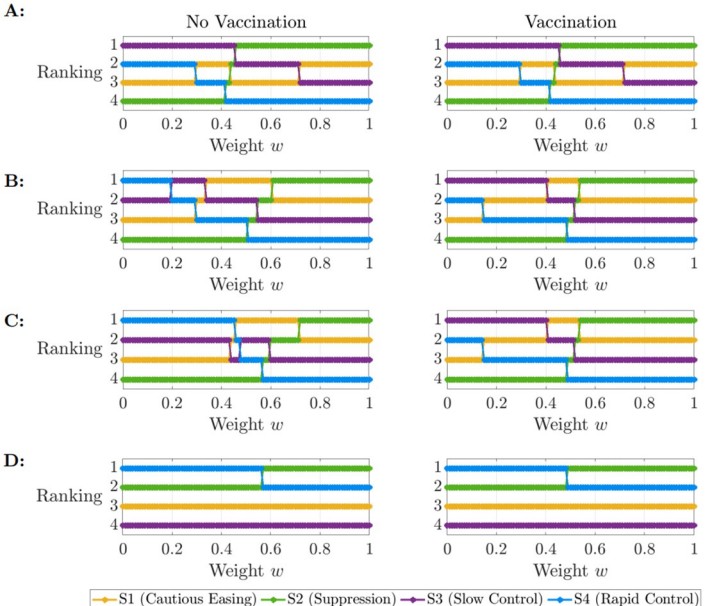

**Fig 2. Strategy rankings.** Ranking of strategies with respect to weighting $w$ for choice of hospital capacity ($H_c$) and time horizon ($t_f$, in days). The figure shows the scenario of no vaccination (left) contrasted with a vaccination scenario for $\eta = 0.9$ and $T = 360$ days (right). Shown are cost outputs for A: $H_c = 1,250$, $t_f = 360$, B: $H_c = 1,250$, $t_f = 720$, C: $H_c = 1,250$, $t_f = 1,080$ and D: $H_c = 1,000$, $t_f = 1,080$.

Fig 3C). In contrast, the Slow control and Rapid control strategies are optimal for lower values of $w$, with the optimal strategy in that case depending on the time at which the vaccine will be developed and the eventual vaccination coverage.

The probability distributions which characterise vaccine-related uncertainty (detailed in S2 Fig) were used to generate a distribution of possible outbreak costs under the four strategies, assuming that $H_c = 1,250$ individuals. These costs are shown in Fig 4, centre. Predictably, a shorter waiting time $T$ for vaccine development and a greater eventual coverage $\eta$ were associated with the smallest expected cost; both of these aspects are necessary to achieve a significant reduction in cost in comparison to the pessimistic distribution (Fig 4C). The expected cost of the Suppression strategy decreased with $w$ and increased for the remaining strategies. The Suppression strategy generally had the widest credible interval for cost. While this strategy certainly benefits from a prompt vaccine arrival, delays to vaccine deployment resulted in overwhelming costs to maintain interventions with minimal reduction in hospitalisation as alternative strategies returned to low levels with less prolonged interventions.

For each distribution characterising uncertainty in when a vaccine will be developed and its eventual coverage, we calculated the probability that each strategy leads to the lowest cost (Fig 4, right) or the highest cost (S7 Fig). These probabilities were calculated by first ranking the strategies separately for each combination of $(T, \eta)$. Then, the proportion of values (weighted by the relative likelihood of each $(T, \eta)$ combination) in which each strategy leads to the lower (or highest) cost was computed. Conveying strategy performances in terms of probabilities can assist the decision making process, as policy makers may be inclined to adopt strategies with a high likelihood of being optimal or a low likelihood of performing poorly. Regardless of the underlying vaccine distribution, the Suppression strategy had a high probability of being optimal for larger $w$, but the performance of the Slow control strategy for low $w$ depended upon the probability distribution characterising future vaccine availability and coverage (i.e.,

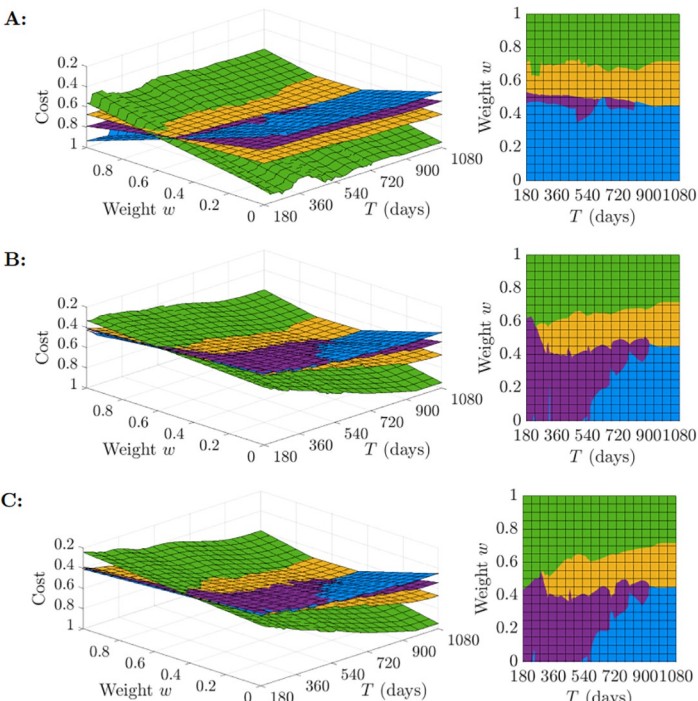

**Fig 3. Outbreak costs under different strategies for different assumptions about the timing of vaccine development and the eventual vaccination coverage.** Surface plots of cost (evaluated by the objective function (1) for different values of the weighting $w$ and the timing of the arrival of the vaccine (left). The z-axis has the lowest cost at the top, allowing the optimal strategy (corresponding to the lowest cost) to be seen more clearly. Results were generated separately for vaccine coverage A: $\eta = 0.1$, B: $\eta = 0.5$ and C: $\eta = 0.9$. The strategies are coloured as follows: yellow (Cautious easing), green (Suppression), purple (Slow control) and blue (Rapid control).

this strategy performed well when the vaccine was expected to be developed quickly and when it was expected to have high coverage). Since the Suppression and Rapid control strategies typically resulted in the largest level of disease control or hospitalisations respectively, they also maximised the probability of having the highest cost irrespective of the vaccine probability distribution when the weighting $w$ was unfavourable (see S7 Fig).

We also evaluated the strategies based upon different ranking criteria (see Methods). For the same weight $w$, the optimal choice was dependent upon the ranking criteria used (S8 Fig). For the Optimistic joint probability distribution (Fig 4B), the Suppression strategy was favourable for intermediate $w$ when the goal is to minimise the expected (mean) cost, most likely cost (mode of the vaccine distribution) or maximise the probability of having the lowest cost. However, it did not fare as well if policy makers instead choose to minimise the 95th percentile cost. Likewise, the performance of the Slow control strategy worsened if we consider the 95th percentile cost in the case of lower $w$. This again highlights the need for policy makers to consider their objectives carefully when determining the optimal strategy to implement.

## Discussion

In this study, we have demonstrated that optimal interventions in the early stages of infectious disease epidemics depend on the precise objectives of the policy maker. The optimal strategy to introduce depends on the extent to which the policy maker chooses to prioritise lowering negative disease-related costs (e.g., numbers of hospitalisations) or limiting the costs of

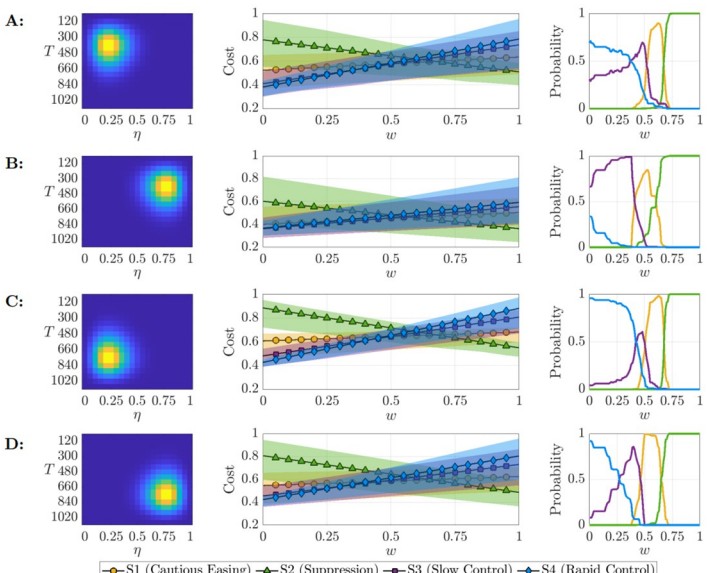

**Fig 4. Effect of uncertainty in when a future vaccine will be developed and its likely coverage on the outbreak cost and the optimal strategy.** The underlying distribution of vaccine outcomes (left), resulting in cost distributions for the control strategies which varied according to weight $w$ (center), in addition to the probability each strategy had the lowest cost across outcomes (right). The joint distributions (rows) covered a range of optimistic and pessimistic scenarios regarding the effectiveness of the vaccination campaign; A: D1 (Radical) distribution, B: D2 (Optimistic) distribution, C: D3 (Pessimistic) distribution, D: D4 (Conservative) distribution. The expected costs, $\mathbb{E}[J]$, are marked by the scatter points, and the shaded regions indicate the 2.5th and 97.5th cost percentile.

maintaining interventions (e.g., the economic and social costs associated with stringent measures)—see Fig 2. We also showed how key uncertainties can be accounted for in the decision making process—for example, uncertainty in the time required for a vaccine to be developed and the level of vaccine uptake in the host population (Fig 4).

Of the strategies that we considered, we found that the stringent Suppression strategy was most likely to be optimal if a vaccine was expected to be developed quickly and a high vaccination coverage could be achieved (Fig 3C). In that scenario, intensive interventions could limit the negative effects of disease until the vaccine became available. In contrast, less stringent NPIs were more likely to be beneficial if vaccine development was expected to take a long time and/or vaccination uptake was expected to be low (Fig 3A). In addition, we found that the optimal strategy can differ according to whether a policy maker prefers to minimise the expected outbreak cost or the risk of a "reasonable worst case" scenario occurring (S8 Fig).

The work that we have presented here builds on a substantial body of previous research. The objective function used to evaluate strategies for disease control in this study was informed by cost functions that have previously been used when optimal control theory approaches have been applied in the context of infectious disease epidemiology [12–16]. The nature of these functions allow for the dual consideration of health-associated costs in addition to control-associated costs when deciding upon an optimal strategy. Previous research articles have investigated the impacts of vaccination in the context of the COVID-19 pandemic using scenario modelling, including considering different vaccine effectiveness and uptake levels [19], deployment strategies [5, 24] and assumptions surrounding the health and economic benefits of vaccination [20]. Modelling of this type is increasingly used in real-time during infectious disease outbreaks to guide policy decisions. Our study extends this previous work by considering how

uncertainty relating to future vaccination (i.e., the time required to develop the vaccine and its effectiveness or uptake) can be incorporated into methods for determining optimal interventions in the initial stages of an infectious disease outbreak.

As with any mathematical modelling study, our research involved necessary simplifications. The objective function used was relatively simple, allowing us to consider how disease-related costs could be balanced against the costs of disease control in as straightforward a setting as possible. However, the overall cost of an epidemic is not limited to the factors that we considered (i.e., hospitalisations and the maintenance of interventions). Accurate assessment of the cost of an outbreak should require other factors to be included and quantified, for example long-term health harms due to stringent interventions being implemented for long periods. This can include mental health impacts [25] or complications arising from the cancellation of elective surgeries [26]. The true costs incurred as a result of an epidemic can be extremely complex to measure and may continue to accumulate beyond the lifetime of the epidemic. Previous work has investigated the economic cost of implementing interventions by fitting polynomial functions describing GDP reduction as a function of lockdown intensity and duration [27], or by modelling GDP reduction as a function of disease and unemployment [28]. In this study, we imposed a constraint on the minimum duration spent in each control state to reflect the fact that policy makers are typically unwilling to change public health measures multiple times in quick succession. However, the act of implementing or removing an intervention may come with a quantifiable cost. This could be incorporated into our modelling framework by adapting the current objective function, to penalise policies that switch between control states frequently. Such an extension could take the form of a sunk cost for switching into a new control state, or a cost that decays as a function of time between policy changes.

Another potential future consideration is that the effectiveness of different measures may change during an epidemic—for example, lockdown may be less effective if it is introduced multiple times or maintained for a long period, since this would likely lead to a reduction in public compliance with government guidelines. Currently, our modelling framework does not account for spontaneous behaviour change in the population in response to the perceived risk of infection, and behavioural responses to interventions are not modelled explicitly. In practice, individuals can exhibit prevalence-elastic behaviour, i.e. the voluntary adoption of measures that limit transmission in response to an increase in the perceived infection risk as the disease becomes more prevalent [29]. On the other hand, the degree of non-compliance with interventions in the population can change during an epidemic, particularly when measures have been introduced and relaxed several times. Such changes in population behaviour have a direct consequence on the infectious disease dynamics and thus the impact of interventions [30, 31]. Anticipating population responses to public health messaging remains a challenge, and public health interventions should be designed while considering potential resulting changes in behaviour [32].

In addition, we considered a limited range of strategies for disease control in this study. All of these strategies were based on thresholds in the prevalence of hospitalised individuals. However, in principle other triggers for changing NPIs could be considered, for example thresholds in numbers of infections or deaths, or in estimated outbreak growth rates or reproduction numbers. Further investigation into the impacts of different approaches for determining when interventions should be changed is a key area for future work. Even if thresholds in the numbers of hospitalised individuals are used to determine when to amend interventions, in principle the values of the thresholds could be optimised considering all possible values (as opposed to the four strategies that we considered here), at the cost of a huge range of different viable strategies to explore. We also acknowledge that the predicted effectiveness of different interventions will depend on the underlying model used to simulate their impacts. The model

structure used in this study was motivated by compartmental models used during the COVID-19 pandemic [33], but other model architectures and settings are possible. Before mathematical models can be used to inform policy, one should carefully consider (i) the appropriate model structure to match the epidemiology of the pathogen driving the outbreak under consideration and (ii) the precise control strategies to consider (limiting this choice to those that can be applied in practice). Extending our approach to consider a range of different compartmental models suitable for modelling a range of pathogens is an area for future research.

To limit the dimensions of uncertainty, the vaccine in this study was only allowed to vary in its deployment time $T$ and eventual coverage $\eta$. Here we use the parameter $\eta$ as a catalyst to capture a wide array of different vaccine-induced dynamics which range from minimal effect to driving the pathogen extinct. In reality, the efficacy of vaccines against infection, disease and hospitalisation can be unknown prior to large-scale clinical trials and deployment. Increasing the complexity of vaccination can be examined in future projects. Epidemiological parameters in the disease model were fixed as the primary objective was to investigate vaccine-induced uncertainty, however at the beginning of a novel outbreak there can be substantial uncertainty regarding pathogen transmission and disease severity. Sensitivity analyses suggest that despite noticeable changes in epidemic trajectories for the four strategies as parameters are varied, their relative ranking in cost contributions remains largely unchanging (see supplementary material S1 Text). Scenario modelling is always sensitive to parameterisation and this motivates the resolution of parameter uncertainty as early as possible during infectious disease outbreaks.

Nonetheless, despite these simplifications, our approach enabled us to demonstrate clearly that identification of the optimal strategy relies upon the objectives of disease control being set out clearly by policy makers. If these objectives are clearly stated, and if relevant costs can be quantified, then epidemiological modelling can be used to assess the effectiveness of different strategies. Moreover, in settings in which there are relevant uncertainties—for example, uncertainty in the precise costs of different strategies or uncertainty about when a new vaccine may be developed—then the modelling framework that we have presented can be used to identify the optimal strategy accounting for those uncertainties (i.e., based on the best available evidence). As we showed by including the potential for a vaccine to be developed, our modelling framework is easily extensible. We therefore hope that this research lays the foundation for policy makers to be able to assess the likely effectiveness of different strategies during future outbreaks of a range of different pathogens.

## Materials and methods

### Epidemiological Model

We consider a general deterministic model of pathogen transmission in a closed population, comprising a system of ordinary differential equations (see supplementary material S1 Text). It is based upon the model initially developed by Keeling *et al.* [33] to simulate COVID-19 dynamics within the United Kingdom, although as noted above explicitly modelling COVID-19 is not the focus of our study. The population comprises of individuals subdivided into three demographic cohorts of ages $G = \{0 - 19, 20 - 64, 65+\}$ to reflect variability in disease severity with age; namely, the age-dependent probabilities of developing symptomatic infection ($d_a$) and the probabilities of hospitalisation given symptomatic infection ($h_a$). Transmission is also dependent on the ages of the potential infectors and infectees, based upon mixing matrices derived from social contact patterns in the United Kingdom [34]. Transmission rates are scaled so that the basic reproduction number (calculated using the Next Generation Matrix approach [35]) takes the value $R_0 = 3$.

The model includes four infectious classes. Asymptomatic infection is differentiated from symptomatic infection, which is further subdivided by severity (including whether the individual is hospitalised). The prevalence of individuals in the hospitalised compartment is used as a trigger to introduce or relax interventions. All four infectious classes contribute towards transmission, although we assume that asymptomatic individuals are less infectious (e.g., due to a lower viral load) and hospitalised individuals have significantly fewer contacts due to ward isolation (while allowing for the possibility of nosocomial transmission [36]). The waning of immunity is possible sometime after an individual recovers from infection. The time spent in the recovered class is assumed to follow an Erlang distribution, so that immunity is unlikely to wane immediately (as opposed to under the more standard assumption of an exponential distribution). The mean of this distribution is 800 days and the standard deviation is 462 days. This broad distribution was chosen to allow reinfections to occur over the timescale of the simulated outbreaks, but to ensure that infection induces immunity in most infected individuals in the initial stages of the outbreak. We also take deaths in hospital settings into consideration.

Gamma or Erlang distributions have been found to represent epidemiological periods more accurately than exponential distributions [37]. The Erlang distributed period of waning immunity is implemented using the method of stages [37], which is also used to obtain Erlang distributed latent and infectious periods in our model. A schematic illustrating the model's compartmental structure is shown in Fig 5. A detailed description of the model is provided in the supplementary material S1 Text.

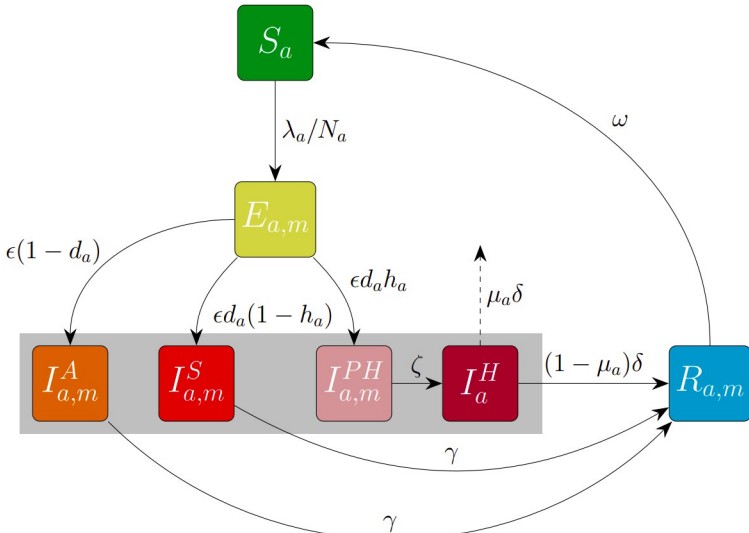

**Fig 5. Epidemiological model structure.** Schematic illustrating the structure of the model that governs the disease dynamics. Susceptible individuals ($S_a$) can acquire infection from any of the four infectious classes (asymptomatic ($I_{a,m}^A$), symptomatic without requiring hospital treatment ($I_{a,m}^S$), symptomatic and eventually requiring hospital treatment ($I_{a,m}^{PH}$) and hospitalised with severe disease ($I_a^H$)), highlighted by the grey box. Immediately following infection, individuals temporarily occupy an exposed class ($E_{a,m}$) where they spend the latent period, before progressing to the infectious classes according to age-dependent severity parameters $d_a$ and $h_a$. At the end of the infectious period, individuals who recover remain in the $R_{a,m}$ class until they undergo natural waning immunity and return to the susceptible class. Disease-induced mortality occurs in hospital settings at age-dependent probability $\mu_a$, and deceased individuals are removed from the model. Where relevant, subscripts distinguish between demographic cohorts $a \in G = \{0-19, 20-64, 65+\}$ and Erlang-stage classification $m \in \{1, 2, 3\}$.

## Disease Control

Policy makers are assumed to manage the spread of the pathogen by switching between three control states for population-wide physical distancing measures. The policy makers can impose Inaction (state 0) on the system or implement interventions which vary between Partial Lockdown (state 1) or Lockdown (state 2). Each control state is associated with a fixed reduction in all transmission rates $\beta_{ij}$ (40% and 70% reductions in transmission respectively for the Partial Lockdown and Lockdown states in comparison to the Inaction state). Switching between states is determined by relaxation or reintroduction of measures based upon the total active number of individuals in the hospitalised compartments, $I^H(t) = \sum_{a \in G} I_a^H(t)$, in the simulation. We incorporate different policies by defining four unique strategies with distinct relaxation and reintroduction thresholds. We note that the thresholds for introducing and relaxing interventions are not necessarily identical. For example, policy makers may relax measures from Lockdown to Partial Lockdown at a lower threshold number of hospitalisations than are required to move from Partial Lockdown to Lockdown, reflecting the fact that measures are only likely to be relaxed when there is substantial evidence that the outbreak is being controlled effectively.

The principles of each strategy that we consider are summarised below.

- **S1 (Cautious easing):** This strategy sets low thresholds for relaxing interventions with a high threshold for entering Lockdown. This strategy is conservative in that it aims to enforce moderate disease control throughout the epidemic but simultaneously seeks to limit the number of phases and duration spent in lockdown.

- **S2 (Suppression):** This strategy aims to maintain a low prevalence, using very low thresholds for relaxation and introduction, at the cost of frequent jumps between Lockdown and Partial Lockdown. This is an extreme case which prolongs the epidemic but may be beneficial when the arrival of an effective vaccine is likely.

- **S3 (Slow control):** Similarly to the Cautious easing strategy, this strategy enforces low thresholds to return to Inaction and a high threshold to enter Lockdown. In contrast, the threshold to relax from Lockdown to the Partial Lockdown state is high, in the hope that relaxing to the Partial Lockdown state is sufficient to prevent disease resurgence.

- **S4 (Rapid control):** This strategy is characterised by a high switching threshold to return to Inaction and a narrow gap between all thresholds. It allows for swift relaxation from Lockdown to Inaction, keeping the duration of continuous intervention periods low. Prevalence of infections and hospitalisations can be higher than other strategies, but measures can be reintroduced swiftly to avoid exceeding healthcare capacity.

The principles of each strategy are encapsulated by four switching thresholds—two each for relaxation and reintroduction of interventions. We use $T_{ij}$ to depict the switching threshold (in the number of currently hospitalised individuals) for moving from control state $i$ to control state $j$. Table 1 lists the switching thresholds for the four strategies considered. Decisions regarding the switching of interventions are discrete and occur at the beginning of each day in the simulation. The new control state is implemented after a three-day delay following the switching threshold being triggered to account for logistics and the change in behaviour of the population in response to new interventions. Control states must last at least three weeks before relaxation can take place and at least two weeks before reintroduction can be imposed, as in reality policy makers are unable to change policy on a daily basis. This is inclusive of the aforementioned three-day implementation delay. If the outbreak is rapidly growing and the $T_{12}$ threshold is triggered during the Inaction state, policy makers move directly to Lockdown.

**Table 1. Strategy thresholds for hospital occupancy.** Thresholds $T_{ij}$ for the number of active hospitalisations $I^H(t)$ required to switch from control state $i$ to control state $j$ under each of the four strategies. Control states are denoted as 0: Inaction, 1: Partial Lockdown and 2: Lockdown in the subscripts for thresholds $T_{ij}$.

| Strategy ID | Name | $T_{01}$ | $T_{12}$ | $T_{21}$ | $T_{10}$ |
|---|---|---|---|---|---|
| S1 | Cautious easing | 100 | 500 | 150 | 50 |
| S2 | Suppression | 100 | 200 | 150 | 50 |
| S3 | Slow control | 200 | 500 | 400 | 100 |
| S4 | Rapid control | 300 | 450 | 350 | 200 |

However, it is not possible to skip the Partial Lockdown state for total relaxation; we assume that Partial Lockdown is a necessary precursor to returning to Inaction. Policy makers will also never choose to relax measures if hospital occupancy is growing at the time that the relevant relaxation threshold is triggered.

## Objective function

We define $H(t)$ and $u(t)$ to be the number of new hospital admissions and the level of disease control (relative reduction in transmission), respectively, over the previous day $[t - 1, t]$ in the epidemic, for $t \in \{t_0 + 1, \ldots, t_f\}$. The objective function $J$ that we use in this study to evaluate epidemic costs given a strategy takes the following form:

$$J(H, u, w) = \sum_{t=t_0+1}^{t_f} \left( w \frac{H(t)}{\phi_b} + (1 - w) \frac{(u(t))^2}{\phi_s} \right)$$
$$+ \exp(w_H(\max[I^H(t)] - H_c)). \tag{1}$$

The cost function contains three major components each associated with weights ($w$, $w_H$). The first component measures severe disease outcomes based on total hospital admissions. The second component measures the cost of maintaining interventions which may be associated with a continuous economic cost. We assume this to scale non-linearly with greater levels of stringency, so that Lockdown is disproportionately costly. The constants $\phi_b = 30, 100$ and $\phi_s = 252.5$ normalise both cost contributions against the highest costs observed across all scenarios explored in this work, so that each cost contribution varies on a similar scale. The weighting $w \in [0, 1]$ represents the extent to which the policy maker trades-off the competing objectives (increasing $w$ places a larger emphasis on reducing the direct public health costs as opposed to the costs of interventions). The third (exponential) component acts to heavily penalise strategies under which a specified capacity on peak hospitalised individuals ($H_c$) is exceeded. The rate at which strategies with epidemic peaks beyond or near capacity are penalised is controlled by the parameter $w_H$. This component gives a negligible contribution to strategies which maintain a below-capacity level of hospitalised individuals, but a rapidly increasing contribution once hospital capacity is exceeded (See S3 Fig for a demonstration of this qualitative behaviour). In this study, this weighting is fixed at $w_H = 2$. No discounting of costs is applied in the objective function in our baseline analyses due to the short simulation timescales, but the results were robust to discounting at a rate of 3.5% per annum (S9 Fig).

This objective function can be used to evaluate a standalone cost for a given scenario or yield a distribution of costs when uncertainty is incorporated. It then becomes important to consider the summary statistic used to identify the optimal strategy which may reflect a level of acceptable risk. A robust strategy could require, for example, a low risk of a reasonable worst-case scenario in addition to a low expected cost. Thus, different statistics arising from the

distribution of strategy costs were calculated to explore how the effectiveness of the four strategies are affected by uncertainty.

## Vaccination

In the midst of a global pandemic, vaccination rates can be nonlinear due to logistical bottlenecks and global inequalities [38]. Sasanami *et al.* [39] previously used a logistic function to model COVID-19 vaccine coverage in Japan. In a similar vein, in our work the number of cumulative vaccinations of age cohort $a \in G$ at a time $t$ days into the simulation—after the specified arrival date $T$ of a vaccine—is calculated using the logistic function,

$$V_a(t) = \frac{\eta N_a}{1 + \exp(-\kappa(t - (T + t_c)))}. \tag{2}$$

The parameter $\eta \in [0, 1]$ encapsulates the efficacy and eventual coverage of the vaccine. Given the nature of our model, we are not explicitly considering vaccine efficacy and vaccine coverage independently and therefore $\eta = 0.5$ can be interpreted as either a vaccine with 50% eventual coverage with 100% efficacy against infection, or a vaccine with a higher coverage and lower efficacy against infection (resulting in an overall 50% reduction in susceptibility). Henceforth, we refer to $\eta$ as the eventual coverage of the vaccine. Adjusting the eventual coverage $\eta$ can reflect vaccination campaigns with varying success. For fixed $\eta$, the remaining parameters $\kappa$ and $t_c$ control the speed of increase of the vaccine rollout and the duration required (measured from deployment date $T$) to obtain 50% of the final coverage, respectively.

Vaccination follows an age-staggered protocol by beginning with the 65+ demographic on deployment date $T$, before commencement of the $20 - 64$ and $0 - 19$ age groups after a delay of $t_c/2$ and $t_c$ days, respectively. Vaccination is assumed to target individuals in the susceptible and recovered classes. S4 Fig demonstrates the cumulative vaccination uptake using the logistic function (2) for an example simulation, using an arrival date of $T = 360$ days and parameters $\eta = 0.9$, $\kappa = 0.05$, $t_c = 100$. The parameters controlling the shape of the logistic function (namely, $\kappa$ and $t_c$) are fixed and the uncertainty in strategy cost is generated by varying the arrival date $T$ and coverage $\eta$.

We incorporate uncertainty in the vaccine arrival time and coverage by selecting a specified arrival date $T$ and parameter $\eta$ from discrete probability distributions, to limit the number of required model simulations. We allow $T$ to take discrete values between 60 and 1,080 days, in increments of 60 days. Similarly, we allow $\eta$ to take discrete values between 0 and 1, in increments of 0.05. Four discrete probability distributions are generated to reflect distinct levels of optimism regarding the timing and eventual coverage of the future vaccine. Visualisation and details regarding the generation of these distributions are provided in the supplementary material S2 Text. The principle behind each distribution considered is summarised in the list below.

- **D1 (Radical):** Timeliness is prioritised over maximising efficacy in order to enable vaccination rollout as early as possible. Characterised by $\mathbb{E}[T] = 420$ days, $\mathbb{E}[\eta] = 0.25$.

- **D2 (Optimistic):** The global focus on vaccine development has enabled a highly effective vaccine to be deployed promptly. Characterised by $\mathbb{E}[T] = 420$ days, $\mathbb{E}[\eta] = 0.75$.

- **D3 (Pessimistic):** Global factors render the vaccine significantly less timely and effective in comparison to those developed for COVID-19. Characterised by $\mathbb{E}[T] = 720$ days, $\mathbb{E}[\eta] = 0.25$.

- **D4 (Conservative):** A lengthier timeline for vaccine research and development leads to a higher quality vaccine. Characterised by $\mathbb{E}[T] = 720$ days, $\mathbb{E}[\eta] = 0.75$.

Uncertainty leads to a cost distribution for each strategy conditional on the joint probability distribution for the timing and eventual coverage of the vaccine. Denoting the set of possible combinations of vaccine arrival times and coverage by $\Psi$, the expected cost of a strategy $\mathbb{E}[J]$ is conditional on all plausible outcomes of $T$ and $\eta$,

$$\mathbb{E}[J] = \sum_{(T,\eta)\in\Psi} \mathbb{E}[J|T,\eta]\,\mathbb{P}(T,\eta)\,, \tag{3}$$

where $\mathbb{P}(T,\eta)$ is the probability mass function of the vaccine joint probability distribution. For a distribution of cost, we rank the strategies based upon a number of criteria:

1. Minimise the expected (mean) cost, $\mathbb{E}[J|\text{Strategy}]$.

2. Minimise the cost of the "most likely outcome" (corresponding to the mode of the $(T,\eta)$ joint distribution, disregarding any other potential outcomes).

3. Minimise the 95th percentile cost (the cost $J_c$ such that $\mathbb{P}[J > J_c|\text{Strategy}] \approx 0.05$).

4. Maximise the probability of having the lowest cost, $\sum_{(T,\eta)\in\Psi^S}\mathbb{P}(T,\eta)$, where $\Psi^S \subset \Psi$ is the set of outcomes in which strategy $S$ is optimal (out of the four strategies considered).

Some criteria focus upon minimising the objective function by averaging over all scenarios or considering scenarios that are most likely to happen; for example, minimising the expected cost $\mathbb{E}[J|\text{Strategy}]$ or the cost for the mode of the underlying vaccine distribution. Other criteria can be viewed as a measure of strategy robustness, such as the 95th percentile cost which a policy maker may perceive to reflect a "reasonable worst-case scenario". Such situations are particularly important to consider when there is high uncertainty regarding the future progression of the outbreak, or when extreme scenarios can be particularly detrimental.

## Supporting information

**S1 Text. Epidemiological model.** Contains further information regarding the epidemiological model structure and parameters. Contains additional sensitivity analysis figures.
(PDF)

**S2 Text. Generating vaccine uncertainty.** Clarifies the construction of vaccine marginal and joint probability distributions.
(PDF)

**S1 Fig. Vaccine distributions (marginal).** Marginal probability distributions for the vaccine deployment date $T$ (left) and eventual coverage $\eta$ (right). The distributions are generated using the random variables $Y$, $Z$ who arise from discrete normal distributions with respective parameterisation $(\mu_T, \sigma_T^2)$ and $(\mu_\eta, \sigma_\eta^2)$. Different expectations for $T$, $\eta$ are driven by varying $\mu_T$, $\mu_\eta$ and the underlying variance is fixed, $\sigma_T^2 = \sigma_\eta^2 = 2.4$.
(TIF)

**S2 Fig. Vaccine distributions (joint).** Joint probability distributions for vaccination uncertainty characterised by deployment date $T$ and eventual coverage $\eta$. The joint distributions cover a range of optimistic (top right: low $T$ and high $\eta$) and pessimistic (bottom left: high $T$ and low $\eta$) scenarios regarding the effectiveness of the vaccination campaign.
(TIF)

**S3 Fig. Qualitative behaviour of exponential term in the objective function.** Qualitative behaviour of the exponential term in the objective function (1). Since the public health and control contributions are bounded in [0, 1] (shaded region), any excess on peak hospital

capacity $H_c$ renders the strategy infeasible with a cost exceeding at least one, regardless of weight $w_H$. The weight $w_H$ can be scaled to deal with different perspectives regarding risk to overwhelm, but will be fixed at $w_H = 2$ (red) for this study.
(TIF)

**S4 Fig. Qualitative behaviour of vaccination parameters.** Cumulative vaccinations by age cohort $V_a(t)$ in a chosen simulation with parameter choices $\eta = 0.9$, $\kappa = 0.05$, $t_c = 100$ days and deployment date $T = 360$ days.
(TIF)

**S5 Fig. Hospitalisation dynamics with vaccination.** Number of active hospitalised individuals across all age cohorts $I^H(t)$ for each strategy, with a vaccine introduced on day $T = 360$ with $\eta = 0.9$. The periods spent in the Lockdown state and the Partial Lockdown state are shaded in red and yellow respectively. Blue dashed lines represent the reintroduction thresholds (where $T_{12} > T_{01}$), and blue dotted lines represent the relaxation thresholds (where $T_{21} > T_{10}$). Shaded are the periods spent in (red) Lockdown, (yellow) Partial Lockdown and (white) Inaction. Strategies are labelled as A: S1 (Cautious easing), B: S2 (Suppression), C: S3 (Slow control) and D: S4 (Rapid control).
(TIF)

**S6 Fig. Outbreak costs under different strategies for different assumptions about the timing of vaccine development and the eventual vaccination coverage.** Surface plots of cost (evaluated by the objective function (1)) for different values of the weighting $w$ and the eventual coverage of the vaccine (left). The z-axis has the lowest cost at the top, allowing the optimal strategy (corresponding to the lowest cost) to be seen more clearly. Results were generated separately for vaccine time to deployment A: $T = 360$, B: $T = 630$ and C: $T = 900$ days. The strategies are coloured as follows: yellow (Cautious easing), green (Suppression), purple (Slow control) and blue (Rapid control).
(TIF)

**S7 Fig. Probability each strategy accurs the greatest cost, subject to underlying vaccine distribution.** Probability that each strategy has the greatest cost according to underlying vaccine joint distribution and weight $w$. The joint distributions cover a range of optimistic (top right: low $T$ and high $\eta$) and pessimistic (bottom left: high $T$ and low $\eta$) scenarios regarding the effectiveness of the vaccination campaign. This probability is measured by ranking the strategies across all permutations of timing and coverage parameters $(T, \eta)$ and summing the probabilities where each strategy is the extremum. The strategies are coloured as follows: yellow (Cautious easing), green (Suppression), purple (Slow control) and blue (Rapid control).
(TIF)

**S8 Fig. Varying strategy rankings by criteria.** Strategy rankings according to a range of criteria described in Methods. Given a weighting $w$, the optimal strategy is sensitive to the summary statistic for which the policy maker seeks to minimise the objective function against. The strategies are coloured as follows: yellow (Cautious easing), green (Suppression), purple (Slow control) and blue (Rapid control).
(TIF)

**S9 Fig. Sensitivity to annual discounting.** Surface plots of cost (flipped z-axis) evaluated by the objective function (1) across weighting $w$ and different simulations which vary the arrival date of vaccination (left), and the top surface akin to viewing the optimal strategy for choice of $w$ and vaccine arrival date (right). Simulations were separately generated for vaccine eventual coverage $\eta = 0.9$, where we apply no discounting (A, main text results) and the discounting of

costs at a per-annum rate of 3.5% (B). The strategies are coloured as follows: yellow (Cautious easing), green (Suppression), purple (Slow control) and blue (Rapid control).
(TIF)

## Author Contributions

**Conceptualization:** Nathan J. Doyle, Fergus Cumming, Robin N. Thompson, Michael J. Tildesley.

**Data curation:** Nathan J. Doyle.

**Formal analysis:** Nathan J. Doyle.

**Funding acquisition:** Michael J. Tildesley.

**Investigation:** Nathan J. Doyle.

**Methodology:** Nathan J. Doyle, Robin N. Thompson, Michael J. Tildesley.

**Software:** Nathan J. Doyle.

**Supervision:** Robin N. Thompson, Michael J. Tildesley.

**Validation:** Nathan J. Doyle.

**Visualization:** Nathan J. Doyle.

**Writing – original draft:** Nathan J. Doyle.

**Writing – review & editing:** Nathan J. Doyle, Fergus Cumming, Robin N. Thompson, Michael J. Tildesley.

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
