## [Decision Letter · Decision Letter 0]

2 May 2024

Dear Mr Doyle,

Thank you very much for submitting your manuscript "When should lockdown be implemented? Devising cost-effective strategies for managing epidemics amid vaccine uncertainty" for consideration at PLOS Computational Biology. As with all papers reviewed by the journal, your manuscript was reviewed by members of the editorial board and by several independent reviewers. The reviewers appreciated the attention to an important topic. Based on the reviews, we are likely to accept this manuscript for publication, providing that you modify the manuscript according to the review recommendations.

Sincerely,

Joseph T. Wu

Academic Editor

PLOS Computational Biology

Virginia Pitzer

Section Editor

PLOS Computational Biology

Reviewer's Responses to Questions

**Comments to the Authors:**

Reviewer #1: In this manuscript, Doyle and colleagues present a model-based analysis of the optimal control of an emerging epidemic from the perspective of a central policy planner. They discuss four main policies and assess how the optimal policy changes depending on the planner's beliefs about eventual vaccination coverage and timing. Rather than identifying a single optimal policy, the authors instead highlight how the planner's beliefs and priorities can impact the rankings of the available policies. Policies are assessed not just in terms of their mean expected cost, but also various other statistics including the most likely cost and the robustness (e.g., the likelihood of a worst-case scenario occurring).

The authors have chosen an appropriate model for their analysis, and they have chosen sensible simplifying assumptions (such as reducing the policy space to four main policies and considering four vaccination scenarios that span the space of plausible outcomes). Their conclusions are well supported and the paper as well written overall. My comments are minor.

1. The authors consider the optimal control strategy from the perspective of the central planner, but it is well known from the experience of COVID-19 and other epidemics that individuals also spontaneously change their behavior in response to perceived risk, sometimes reinforcing and sometimes counteracting the policies enacted by a central planner. While formally interrogating the impact of such spontaneous individual behavior is outside the scope of this manuscript, I would encourage the authors to include a section in the discussion pointing this out as a limitation of their model and highlighting some of the relevant literature on risk perception and behavior change in response to infectious disease threats.

2. The authors discuss in the Methods that the central planner is likely to be unwilling to change policies rapidly, and therefore introduced a deterministic time horizon (two weeks) during which policies cannot be changed. It seems that it would be better to include an explicit cost of changing policies -- e.g., a cost that decays as a function of time since the most recent policy change, as an additional term in Eq (1). This way, the rate of policy change would be explicitly included in the cost model. I leave it to the editor and the authors to determine whether this is needed for the paper itself, and if not, I would just encourage the authors to more centrally discuss the fact that changing policies rapidly comes with a cost that can be quantified in a similar manner as the costs already captured by the model.

Reviewer #2: Thank you for the opportunity to review this manuscript. In this study the authors considered mathematical models that are used to help guide public health responses during an outbreak of an infectious disease pathogen. They proposed a compartmental model linked to an objective function, where the objective function balances the costs of control interventions (eg lockdowns), versus the health costs. Different control scenarios are explored, and it is found that the optimal control strategy depends on the extent to which policy makers value reducing health costs, versus reducing the costs of NPIs. This proposed framework could be applied in future epidemic scenarios to help guide and motivate decision-making and clear articulation of priorities and costs.

Main comments

I appreciated that in this study, the authors used the motivation of the COVID-19 pandemic, to then present a modelling framework for informing future decisions that is more broadly applicable beyond just COVID – and I suppose there is a bit of a balancing act between specificity and generalisability here. However I thought that one component that was potentially overlooked here was the model structure itself. To what extent do you think the form of the compartmental age-structured SEIRS-type model selected for this analysis (where the model really applied to COVID) might have changed the results? I’m not suggesting you need to explore other model structures as that would be beyond the scope of the current study but I think worth commenting on how this study is restricted somewhat by the model structure chosen, and that other model structures are possible (e.g. for a vector-born disease where the compartmental structure may look different?).

In the methods, I found the section on “Control” hard to follow – and I think it is to do with the notation and terminology. You explain the three states of “No Control”, “Intermediate Control” and “Lockdown”, and the levels of transmission reduction associated with these (fine). I think these are subsequently referred to as “control states”. But then you talk about “different control policies”, and then “four control strategies” (line 350). As far as I could understand, the control strategies place different thresholds on switching between control states. And then on line 380 you mention “control measures”. I think it is just all the different version of control being used that is a little hard to follow (I had to read it a couple of times to make sure I understood). You may want to consider the terminology more carefully. In Table 1, I also think a bit more detail could be provided in the column headings and the caption.

Minor comments

How were the 40% and 70% levels of reduction in transmission for the control scenarios selected?

I found it really hard to interpret Figure 1 – the switching thresholds indicated by the dashed lines didn’t seem to line up with T_01, T_12, and so on, on the right-hand side axis?

For Figure 2 – what is the difference between the panels in the left column versus the right column?

Line 145-146, where you state that in the model, the value of the weight w relates to the policymakers choice about costs of disesase versus costs of public health measures – would be helpful to explain here what a low versus high value of w means to aid intuition for the reader.

Objective function. Line 401 onwards – did you explain here what w_H is?

Figures S10 to S15 – it’s not clear what values of the parameters being varied are applied in each of the panels within these figures.

Line 60. “when designing models for use to guide interventions”

**Have the authors made all data and (if applicable) computational code underlying the findings in their manuscript fully available?**

Reviewer #1: Yes

Reviewer #2: Yes

PLOS authors have the option to publish the peer review history of their article (what does this mean?). If published, this will include your full peer review and any attached files.

Reviewer #1: No

Reviewer #2: No

Figure Files:

Data Requirements:

Reproducibility:

References:

---

## [Decision Letter · Decision Letter 1]

27 Jun 2024

Dear Mr Doyle,

We are pleased to inform you that your manuscript 'When should lockdown be implemented? Devising cost-effective strategies for managing epidemics amid vaccine uncertainty' has been provisionally accepted for publication in PLOS Computational Biology.

Best regards,

Joseph T. Wu

Academic Editor

PLOS Computational Biology

Virginia Pitzer

Section Editor

PLOS Computational Biology

Reviewer's Responses to Questions

**Comments to the Authors:**

Reviewer #2: Thank you for the opportunity to review this revised manuscript. I am satisfied the authors have addressed my comments and suggestions. I have no further comments.

**Have the authors made all data and (if applicable) computational code underlying the findings in their manuscript fully available?**

Reviewer #2: None

PLOS authors have the option to publish the peer review history of their article (what does this mean?). If published, this will include your full peer review and any attached files.

Reviewer #2: No

---

## [Editor Report · Acceptance letter]

14 Jul 2024

PCOMPBIOL-D-24-00478R1 

When should lockdown be implemented? Devising cost-effective strategies for managing epidemics amid vaccine uncertainty

Dear Dr Doyle,

I am pleased to inform you that your manuscript has been formally accepted for publication in PLOS Computational Biology. Your manuscript is now with our production department and you will be notified of the publication date in due course.

With kind regards,

Jazmin Toth
